# Surgical Treatment of Lung Abscess Due to an Awn Aspiration in a 9-Year-Old Child: A Case Report

**DOI:** 10.3390/children10060910

**Published:** 2023-05-23

**Authors:** Angelina Vlahova, Zdravka Antonova, Edmond Rangelov, Nikola Kartulev, Velichka Oparanova, Natalia Gabrovska, Albena Spasova, Svetlana Velizarova, Hristo Shivachev

**Affiliations:** 1Department of Pediatric Surgery, University Multiprofile Hospital for Active Treatment and Emergency Medicine “N.I.Pirogov”, 1606 Sofia, Bulgaria; 2Department of Pediatrics, Medical University—Sofia, Specialized Hospital for Active Treatment of Children’s Diseases “Prof. Ivan Mitev”, 1606 Sofia, Bulgaria

**Keywords:** awn, grass inflorescence, aspiration, abscess, lung resection, bronchoscopy, child

## Abstract

Introduction: Foreign body aspiration is a common condition in the child population and is one of the leading causes of accidental deaths in children. The aspiration of an awn (grass inflorescences) is extremely rare. Aim of the study: This study aims to describe the symptoms, diagnosis, therapeutic difficulties, and results of the aspiration of grass inflorescence. They are all related to the shape of the awn’s head and its behavior in the tracheobronchial tree. Case description: We present a 9-year-old boy with a history of an awn aspiration and recurrent respiratory infections. After antibiotic and symptomatic treatment, two bronchoscopies were performed, and both showed stenosis and obturation of the segmental and subsegmental bronchi of the left posterior basal segment, but no foreign body was found. After recanalization and continuous medicamentous treatment, a computed tomography (CT) verified the lung abscess. Video-assisted thoracoscopic surgery (VATS) was performed, and an atypical lung resection was conducted. During the surgery, a foreign body—grass inflorescence from the species Hordeum murinum—was found in the resected tissue. The patient recovered uneventfully after the surgery. Conclusions: Grass inflorescence aspiration in the child population is an extremely rare event, and it represents a diagnostic and therapeutic issue. The exact timing of surgery is very important to reduce complications and avoid death.

## 1. Introduction

Foreign body aspiration is a common condition in the child population, and it is one of the leading causes of accidental death in children. According to the literature, it happens most commonly in the first 3 years of life, but a second peak is seen between 10 and 11 years [1]. Aspiration of an awn (grass inflorescences) is an extremely rare event, with potential diagnostic difficulties resulting from its shape [2]. There are two different types of awn head structures: the “lodging” type and the “extrusive” type [2,3,4,5]. In the current presented case, an “extrusive” type was found—a head of barley grass from the species Hordeum murinum, family Poaceae. It is a widespread species in the fields of Europe, including Bulgaria [3].

The medical history of the patient is very important in these cases, but it is not always presented. Diagnostic workup comprises frontal and lateral chest X-ray (verifying signs of pneumonia, atelectasis, pneumothorax, mediastinal shift, and air trapping), computed tomography (CT), and rigid or flexible bronchoscopy [1,2]. In most of the reported cases in the literature, surgery is needed to prevent serious complications. The most common are bronchial obstruction, spontaneous tension pneumothorax, pneumomediastinum, chest wall abscess, pneumonia, pleural effusion, bronchiectasis, bronchopleural fistula, and recurrent hemoptysis [2,6,7].

In this paper, we present the case of a 9-year-old boy with a history of awn aspiration who underwent bronchoscopy and surgery for pulmonary abscess—video-assisted thoracoscopic surgery (VATS) with atypical pulmonary resection. Early bronchoscopy (rigid or flexible) is recommended for precise diagnosis and to prevent complications [4,8,9]. The aim of this study is to describe the symptoms, diagnosis, treatment, and results of the aspiration of grass inflorescence.

## 2. Case Description

We present a 9-year-old boy with a history of recurrent respiratory infections. In July 2022, the child swallowed an awn during a walk on a field in Northwest Bulgaria, and after two days, he started to cough. On the third day after the accident, he became febrile, and antibiotic therapy was prescribed (Clarithromycin). This treatment was not effective, and the child suffered left-sided chest pain, irradiating to his left shoulder. Nonsteroidal anti-inflammatory drugs were prescribed. On the 10th day, the patient was admitted to the Department of Pediatric Pulmonology for further diagnostic workup and treatment. The child’s condition was stable, with a productive cough and reduced vesicular breathing paravertebrally and axillary on the left side of the chest. His hemodynamic was stable, and his abdomen was not tender, without hepatosplenomegaly. His blood tests showed inflammation—increased C-reactive protein (CRP) 36.29 mg/L and leukocytes 10,9 x10^9^ L, and stable hematological parameters (hemoglobin 126 g/L, hematocrit 39%, erythrocytes count range 4.69, platelets 347). Biochemical markers were also in the normal range—sugar level 6.08 mmol/L, creatinine 42 µmol/L, urea 2 mmol/L, ASAT 15 U/L, ALAT 12 U/L, alkaline phosphatase 127 U/L, total protein 65.3 g/L, albumin 41.26 g/L. Frontal and lateral chest X-rays showed faint reticular and patchy opacities with unclear margins in the lower lobe of the left lung (Figure 1A,B). Treatment comprised antibiotics (Ceftriaxone), corticosteroids (Methylprednisolone), and saline inhalations.

The patient’s complaints persisted, and the child was admitted to the Department of Pediatric Thoracic Surgery for further exams. The condition of the child was stable, with reduced productive cough and reduced vesicular breathing paravertebrally and axillary on the left side of the chest. His hemodynamic was stable, and his abdomen was not tender, without hepatosplenomegaly. Blood and biochemical tests were in the normal range, with slightly increased inflammation markers (leucocytes 11.8 G/L, hemoglobin 137 g/L, hematocrit 0.43 L/L, thrombocytes 404 G/L, CRP 0.59 mg/dL, ASAT 21 U/L, ALAT 13 U/L, creatinine 53 µmol/L). Frontal and chest X-rays revealed atelectasis of the posterior basal segment (Figure 2A,B). A flexible bronchoscopy was performed. The trachea, right main bronchus, and segmental and subsegmental bronchi were without obstruction and pathology. The left main and upper lobe bronchi also were normal. A thick secretion was found in the left lower lobe bronchus, and a bronchoalveolar lavage was performed. Stenosis and obturation resulting from hypergranulation were found in the segmental and subsegmental bronchi of the posterior basal segment (LB10). Recanalization of the LB10 was performed with histopathology and microbiology samples. After five days, a second flexible bronchoscopy was made. Poor improvement was marked by persisting stenosis and obturation of LB10. Again, a biopsy and recanalization with new samples were performed. The microbiology was negative, and the histology showed acute bronchitis. Antibiotics (Ceftriaxone) and symptomatic treatment (Pulmicort inhalations) were extended during the entire hospital stay. The patient was discharged on the 7th day and transferred back to the Department of Pediatric Pulmonology for prolonged treatment.

After two weeks, the child was admitted again to the Department of Pediatric Thoracic Surgery. Blood and biochemical tests were in the normal range (leucocytes 8.9 G/L, hemoglobin 136 g/L, hematocrit 0.42 L/L, thrombocytes 281 G/L, CRP 0.06 mg/dL, ASAT 26 U/L, ALAT 17 U/L, creatinine 57 µmol/L). A frontal chest X-ray revealed the persistence of the radiological findings from the last stay (Figure 3). A CT scan showed a zone of consolidation and a typical view of a lung abscess in the left posterior basal segment (Figure 4A,B). Then, the decision for surgery was taken. A left-sided VATS was performed. An abscess (3 cm/4 cm/5 cm) was found in the 10th segment, and an atypical lung resection was performed. After the procedure, the lung specimen was cut, and a foreign body (an awn from the species Hordeum murinum) was found (Figure 5A,B). A 20 Ch drain tube was inserted into the pleural cavity. After the surgery, the improvement was marked. Perioperative and postoperative therapy comprised Sultamicillin, Cefoperazon/Sulbactam, Clindamycin, and postoperative analgesia. The postoperative period was uneventful. The patient was discharged on the 9th postoperative day.

During the follow-up, the child had no complaints, and the physical examination was normal (Figure 6). The incision wounds healed primarily, and no complications were noticed. The patient recovered uneventfully. Annual physical and functional examinations of breathing will be conducted.

## 3. Discussion

Aspiration of grass inflorescence in children is a rare event, and only a few cases have been reported in the literature. For a better understanding of the behavior of such a foreign body, the most important is its shape and progression in the tracheobronchial tree. Awns (also known as grass inflorescences) are characteristic of various plant families, including many types of grass [3]. Grasses (Poaceae) are a family of flowering plants found on fields, roads, or along rivers, and they are widespread in Europe and are typical in many regions of Bulgaria. The well-known barley grass (Hordeum murinum) is one of the most common species, widespread in Europe (Figure 7). Their heads have a special shape with several spikes and bristles, which are responsible for the atypical progression and symptoms of aspiration. Due to this structure, each coughing and respiratory movement helps the migration of an awn towards the periphery of the lung, which can lead to life-threatening complications [2]. Unlike the barley grass, the spikelets of timothy grass (Phleum pratense) are soft and close together. Therefore, they do not penetrate deeply into the lung tissue, and they can be extracted during a bronchoscopy [10]. Usually, the stem of the grass inflorescence goes first into the tracheobronchial tree, and spikelets follow [6]. All kinds of awns are highly allergenic and cause pneumonitis shortly after aspiration [11]. Clinically, there are two types of inflorescences: the “lodging” type, which remains in the bronchus, and the “extrusive type”, which is forced by respiratory movements and migrates toward the periphery of the lung and even through the chest wall. The spikelets of the timothy grass are a typical example of the first type, while the awns of the barley grass are typical for the second type [2,4,5].

The diagnosis is difficult, first because of the young age of the patient (the history of aspiration is not always presented), and second because of the behavior of the grass inflorescence. Some authors noted that an awn inhalation is a seasonal event and occurs in spring and summer (such as in our patient) [5]. The first symptoms are coughing, wheezing, or vomiting, but soon after aspiration, these initial symptoms disappear. A careful medical history must be taken because parents often do not note an episode of choking. Laboratory tests are non-specific, and usually, they show signs of inflammation. Frontal and lateral chest X-rays and ultrasound are strongly recommended to recognize atelectasis, inflammation, or some of the complications of aspiration (pneumothorax, pneumomediastinum, pulmonary abscess, pleural empyema, etc.). Computed tomography is important for the diagnostic workup to identify the foreign body location and to assist in defining the exact timing for surgery [1,2].

Bronchoscopy (rigid or flexible) is strongly recommended for precise diagnosis and to prevent complications [4,8,9]. It is important to know that normal bronchoscopy does not exclude the presence of grass inflorescence, especially when there is an appropriate history [1]. Due to the anatomical structure of the tracheobronchial tree, an aspirated foreign body (including grass heads) is most likely presented in the right main bronchus [2,4]. In cases with a “lodging type” inflorescence, the foreign body can even be extracted during the bronchoscopy. In most cases with an “extrusive type” inflorescence, the foreign body cannot be visualized because it penetrates deeply, and only granulation tissue can be found. In cases of visible extrusive grass inflorescence, extraction is impossible due to the sharp, stiff, and trailing spikes [2,5].

The literature review shows that without surgical treatment, percutaneous elimination through the chest wall may occur. In all cases of awn aspiration except for cases with spontaneous extrusion, surgery is recommended. Thoracotomy with lobectomy is the most frequent [2,4]. Most of the authors prefer to perform an operation as early as possible to prevent complications [8]. The optimal time for surgery is determined by the signs of inflammation seen on the chest X-ray or CT scan [2]. In the presented case, a lung abscess was found, and atypical lung resection was performed.

In all cases with an awn aspiration, at least one serious complication occurs. The most common are bronchial obstruction due to the development of granulation tissue, spontaneous tension pneumothorax, pneumomediastinum, chest wall abscess, pneumonia, pleural effusion, bronchiectasis, and recurrent hemoptysis. Bronchopleural cutaneous fistula can occur when the spikelet extrudes through the chest wall [2,6,7]. According to the literature, the period of time from inhalation to extrusion of the inflorescence through the chest wall is between 10 days and 5 months [5]. Less frequent complications are rib osteomyelitis, brain abscess, liver abscess, or coexisting acute abdomen [2,8]. In this patient’s case, the complication was the development of stricture, obstruction of the LB10, and a lung abscess seen on the CT scan. No other data for pulmonary abscess due to aspiration of grass inflorescence were found in the literature. Unrecognized or late diagnosis of an awn aspiration can cause death.

Suspected differential diagnoses include asthma, diphtheria, pulmonary tuberculosis, chronic bronchitis, bronchiectasis, rib tuberculosis, vascular tumor, and acute appendicitis [4].

In the presented case of a 9-year-old boy with a lung abscess due to an awn aspiration, the decision for operation was made based on the history of the patient, clinical presentation, and findings of X-rays, a CT scan, and a bronchoscopy. The persistence of a lung abscess and failure to find the foreign body during the two bronchoscopies are indications for surgical management. The VATS procedure was chosen because it is less traumatic for the patient, reduces postoperative pain, and reduces hospital stay. Rieth, A. and Clery, A. recommend early surgical management to reduce the complications and risk of death [2,4].

## 4. Conclusions

Awn aspiration is a rare event in the child population. The medical history of the patient is very important in these cases, and it should be taken in detail. The symptoms and clinical and examination findings should be carefully considered. X-ray and CT scan findings of a lung abscess, chest wall abscess, or tension pneumothorax are indications for operation. In cases with suspected awn aspiration, we recommend early radiology examinations and bronchoscopy to reduce complications. The bronchoscopy could be therapeutic if the grass inflorescence is the “lodging” type. In cases with serious suspicion of an awn aspiration, we recommend early surgery (VATS or thoracotomy with segmentectomy or lobectomy). The extent of surgery depends on the location of the foreign body and the inflammation of the pulmonary tissue.

The presented patient had a typical medical history of an awn aspiration with clinical and CT evidence of a lung abscess. Bronchoscopy revealed stenosis and obturation of segmental and subsegmental bronchi of the posterior basal segment (LB10). The foreign body could not be extracted due to its migration into the lung tissue. An operation was performed (VATS with atypical pulmonary resection), and the patient recovered uneventfully. The child will be strictly followed-up—annual physical and functional examinations of breathing will be conducted.

## Figures and Tables

**Figure 1 children-10-00910-f001:**
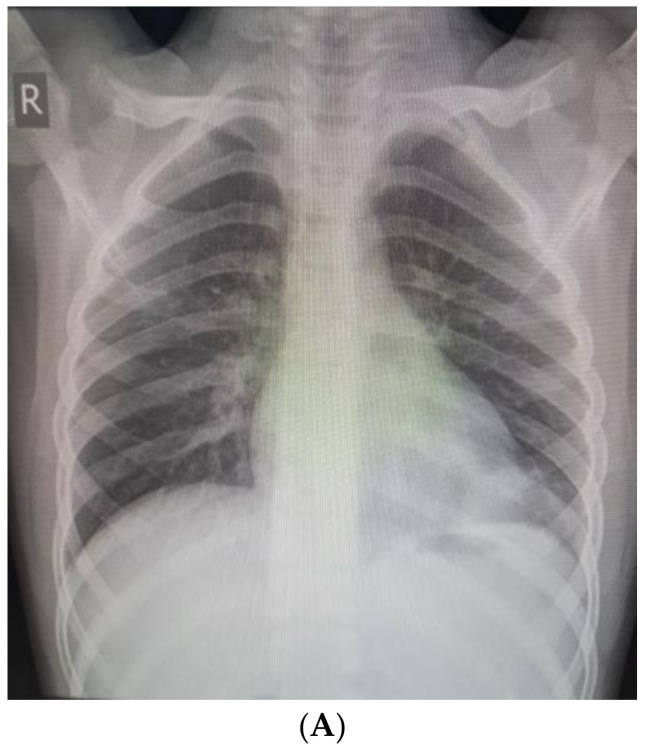
(**A**). Frontal chest X-ray—faint reticular and patchy opacities with unclear margins were visualized in the lower lobe of the left lung. (**B**). Lateral chest X-ray.

**Figure 2 children-10-00910-f002:**
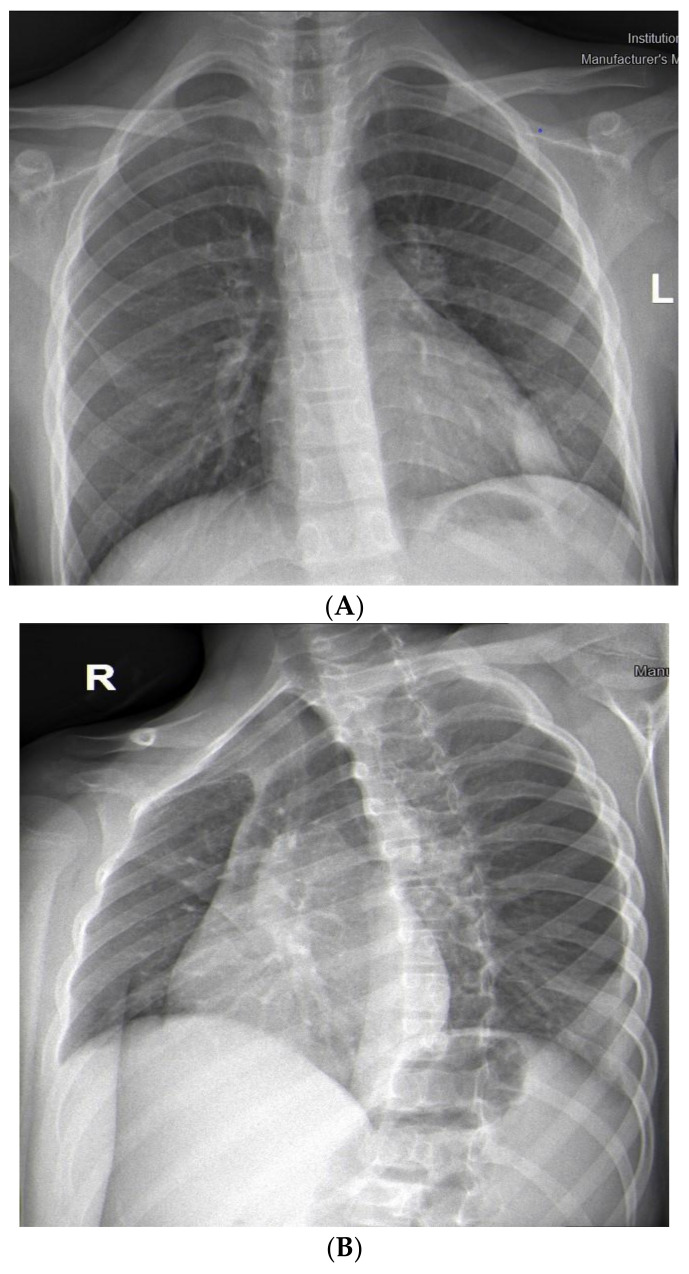
(**A**). Frontal chest X-ray—atelectasis of the posterior basal segment. (**B**). Lateral chest X-ray.

**Figure 3 children-10-00910-f003:**
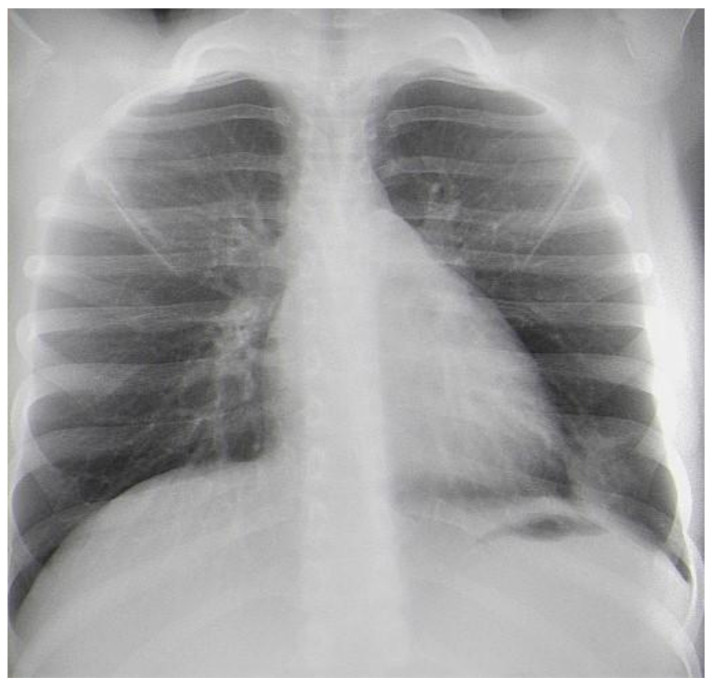
Frontal chest X-ray one day before the surgery.

**Figure 4 children-10-00910-f004:**
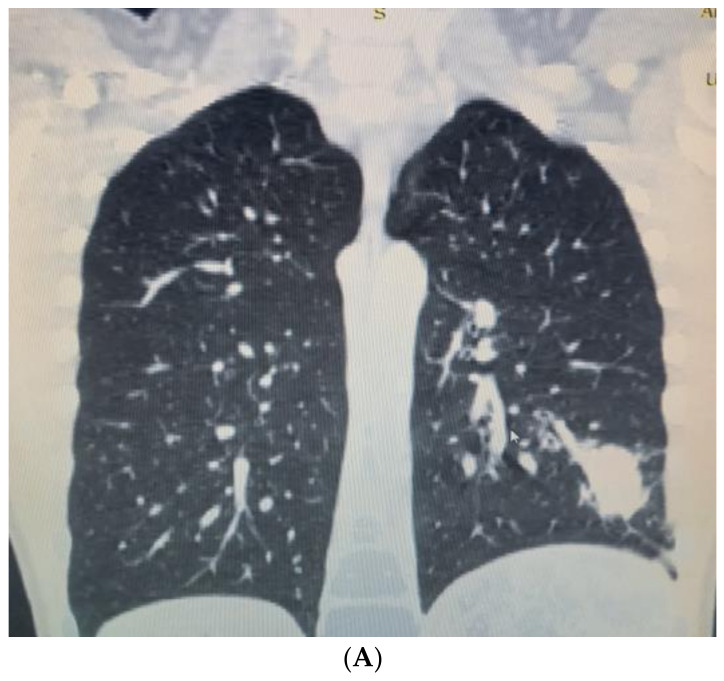
(**A**). CT scan showing lung abscess in left posterior basal segment—a coronal view. (**B**). CT scan—an axial view.

**Figure 5 children-10-00910-f005:**
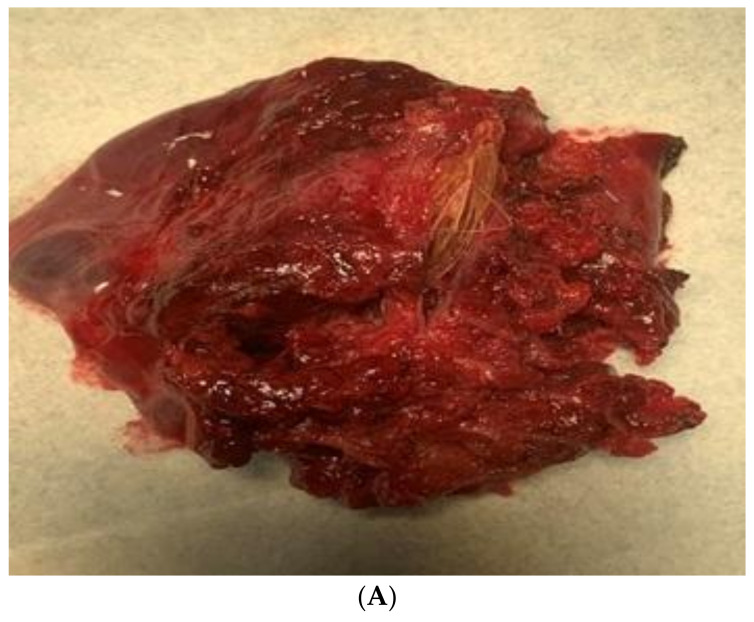
(**A**). A grass inflorescence in the resected lung tissue. (**B**). The length of the awn compared with a scalpel blade.

**Figure 6 children-10-00910-f006:**
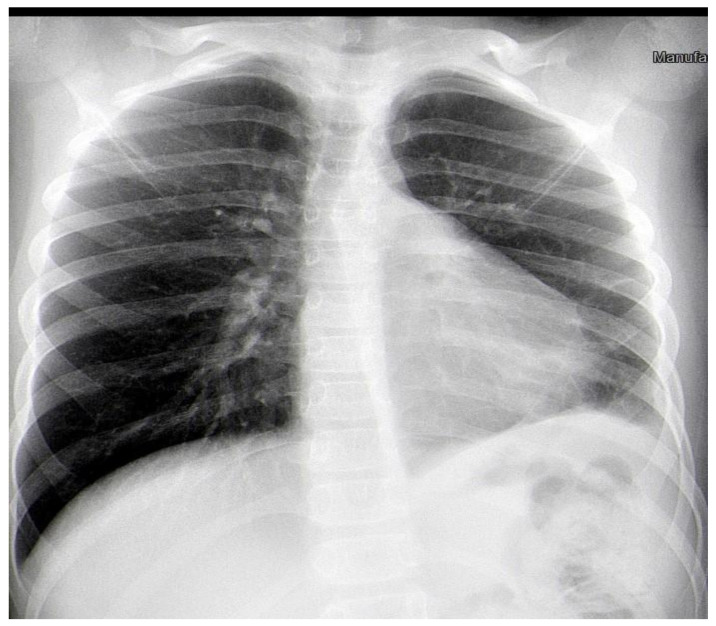
Frontal chest X-ray one month after the surgery.

**Figure 7 children-10-00910-f007:**
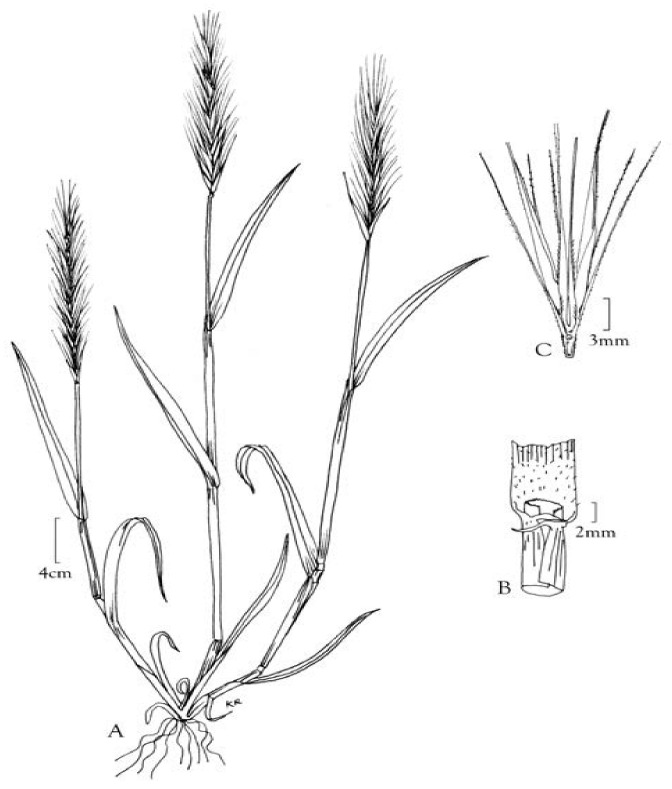
Hordeum murinum. A. Habit. B. Sheath, ligule, and blade. C. Spikelets, adapted with permission from Ref. [12].

## Data Availability

The data presented in this study are available on request from the corresponding author. The data are not publicly available due to privacy restrictions.

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
