# Peer review of "Surgical Treatment of Lung Abscess Due to an Awn Aspiration in a 9-Year-Old Child: A Case Report"

_children, 2023, doi:10.3390/children10060910_

Round 1

Reviewer 1 Report

Dear Authors,

The article makes an interesting contribution to the field on awn aspiration that could cause the lung abscess requiring medical intervention. However, the article needs significant changes in the structure of the text. It needs to put the paper in order, and requires some additions. I have comments before the manuscript could be considered for further proceeding and publication:

Abstract

1.      The abstract is not concise and requires correction in line with the adopted structure for the abstract for the article, for instance: (1) Introduction, (2) Aim of the study, (3) Case description, and (4) Conclusions.

2.      It is not clear, how the foreign body (Hordeum murinum or Phleum pratense) was recognized in this study; and its name should be placed in the abstract.

3.      In addition, sentence in line 23 “We perform a literature review and discuss the diagnostic and management difficulties is inappropriate for an abstract and should be removed.

Introduction

4.      The introduction is non-focused and it lacks a description of the grass that is the subject of this case study; i.e. the systematic name, the species of the plant, origin of the place of occurrence and other characteristics of the grass that are important in the context of swallowed awn by a child.

5.      There is also a need for a literature review and an extension of the research done so far on the health effects of inflorescence aspiration among children. 

Aim of the study

6.      The aim of the study now is „…to discuss the difficulties of the diagnosis, treatment and complications of this rare condition”. I would suggest to change it into “…to describe the symptoms, diagnosis, treatment, and the results of the aspiration of grass inflorescence”.

7.      Please also note that consequently the aim of the study should be identical in the Abstract and in the main text (currently in lines 39-41).

Case description

8.      I would also advise some corrections; for instance, to explain the abbreviations CPR, LB, etc.

9.      More information is needed on the test results of the patient, such as CRP value, leucocytes and other diagnostic tests performed, as well as medications used.

Discussion

10.  Some paragraphs of the article require redrafting by arranging information relating to the results of this case study. Part of the text in lines 108-124 must be in the Introduction.  

11.  I do not fully understand the reasonableness of the Figures 7 and 8 ? Which of the figures presents a foreign body swallowed by child? Please, have the one Figure, which concerns on this case report.

12.  In Discussion section has to be more detailed description, what explains the decisions made in the process of the patient treatment.

13.  It is also important to compare your study with the studies of other authors - are there any other studies supporting your main result?

Conclusions

14.  The sentence, what is in the beginning of the section “Results” “Aspiration of grass inflorescence in children is a rare event and only few cases were reported in the literature”, don’t respond to the aim of the study, so can’t be in the results. Anyway, it could be placed to the Discussion, and expand on this topic, i.e. what has been researched on this topic so far.

15.  The Conclusions of the work should be more informative and response to the purpose of the study. Moreover, the conclusions lack information on the computed tomography used. Please correct them.

General comment

16.  Sources should be cited in the parentheses, for instance, [1,2].

17.  The work has no information in the section Institutional Review Board Statement. Did the study receive the acceptance of Bioethical Committee? Please add the number of this consent and the date.

Please, highlight the changes to the revised version using a different colour.

Author Response

Dear reviewer #1,

Thank you for your constructive remarks and recommendations. In accordance to them, we have made the following corrections and additions to the manuscript (the corrections are in red color).

Abstract

  1. We put in order our abstract in line with the adopted structure for the abstract for the article. We hope it is more comprehensible and concise. 
  2. The foreign body was recognized during the surgery (VATS) and we include this fact in the abstract (currently in line 24). We also include a name of the species Hordeum murinum (currently in line 25).
  3. We remove the sentence “We perform a literature review and discuss the diagnostic and management difficulties”.

Introduction

  1. We add a description of the grass that is the subject of this case study and we also lead in the term “lodging” and “extrusive” types of awns (currently in line 37-41). 
  2. We add the complications induced by the foreign body (currently in line 46-49) and put the literature references. 

Aim of the study

  1. We changed the aim of the study: “…to describe the symptoms, diagnosis, treatment, and the results of the aspiration of grass inflorescence” like you suggested.
  2. The aim of the study is now identical in the Abstract and in the main text.

Case description

  1. We explain the abbreviation of CPR (currently in line 67). The LB10 is explained in the original text. LB10 is the subsegmental bronchi of the posterior basal segment (currently in 89-91) lines. This is a common abbreviation in bronchology. 
  2. We add more information on the test results of the patient and the medications used for the treatment.

Discussion

  1. We constructed the discussion in this manner because we wanted first to explain the structure and specifics of grass inflorescence, second their behavior in the tracheobronchial tree (“lodging” or “extrusive” type) and after that we described the symptoms, diagnosis, treatment and complications. More information about the specific type of the grass inflorescence is put in the introduction.
  2. The reason for Figure 7 and Figure 8 is to show the difference in the shape of the two types of awns. It is important because the shape is determining for the development of complications. We remove Figure 8.
  3. The decision making process is explained in more details in the end of the Discussion (currently lines in 202-208)
  4. We compare the results with other authors. 

Conclusions

  1. The sentence “Aspiration of grass inflorescence in children is a rare event and only few cases were reported in the literature” is moved in the beginning of the Discussion (currently in lines 133-135)
  2. A revision of the Conclusion is made - more information about the patient, CT findings and the purpose of the study is include (currently in lines 210-225)

General comment

  1. Sources are cited in parentheses. 
  2. Ethical statements and Patient consent are attached.

We made the corrections of the inappropriate terms. Thank you very much for your recommendations.

Best regards,

Angelina Vlahova, MD

Reviewer 2 Report

This clinical experience is quite interesting. However, the authors did not include any ethical statement and, in detail, information on the guardians' informed consent to publish the case report. 

Said that, the article itself is not well written and structured. The case presentation should be more ordinated and detailed, also in terms of blood tests, which are not reported at all. 

The discussion should consider if there are any previous and similar reports, in order to include an appropriate literature review. 

Author Response

Dear reviewer #2,

Thank you for your constructive remarks and recommendations. In accordance to them, we have made the following corrections and additions to the manuscript (the corrections are in red color).

Ethical statements and Patient consent are attached to the manuscript.

We made major revisions of our manuscript according to your recommendations - we put more details about the blood tests and therapy.

We also compare our results with other authors at the end of the Discussion.

We made the corrections of the inappropriate terms. Thank you very much for your recommendations.

Best regards,

Angelina Vlahova, MD

Round 2

Reviewer 1 Report

Dear Authors,

Thank you for responding to my previous comments. I believe that the manuscript has improved considerably now and based on this observation, I recommend “Accept”.

Author Response

Thank you very much for your acceptence.

Reviewer 2 Report

Unfortunately, the authors did not improve the manuscript enough, in my opinion. Indeed, although the clinical case itself is interesting, the case report is not well structured and tables providing the full panel of results and, perhaps, a figure showing the longitudinal course should be added.

Moreover, I do not see any case-based review in the discussion (possibly, including a table summarizing the literature search output), although the authors stated in their point-by-point reply that they added changes in the discussion.

Finally, the informed consent was then requested only after the initial submission...anyway, the informed consent is usually in the native language of the patient/guardian.

Also, the English writing needs extensive and professional editing.

Author Response

Dear reviewer #2,

Thank you for your constructive remarks and recommendations. In accordance with them, we have made the following corrections and additions to the manuscript (the corrections are in red color and are marked up using the “Track Changes” function).

We made major revisions of our manuscript according to your recommendations - we made the necessary structural changes, we put more information about the case presentation (currently in line 57 and lines 81-84).

Unfortunately we can not compare our data with other studies because no other studies for lung abscess due to an awn aspiration in the pediatric population were found. We comment this in lines 199-200.

We made the corrections of the inappropriate terms. Thank you very much for your recommendations.

Best regards,

Angelina Vlahova, MD
